# Evaluation of the Hydrolysis Efficiency of Bacterial Cellulose Gel Film after the Liquid Hot Water and Steam Explosion Pretreatments

**DOI:** 10.3390/polym14102032

**Published:** 2022-05-16

**Authors:** Izabela Betlej, Andrzej Antczak, Jan Szadkowski, Michał Drożdżek, Krzysztof Krajewski, Andrzej Radomski, Janusz Zawadzki, Sławomir Borysiak

**Affiliations:** 1Institute of Wood Sciences and Furniture, Warsaw University of Life Sciences—SGGW, 159 Nowoursynowska St., 02-776 Warsaw, Poland; andrzej_antczak@sggw.edu.pl (A.A.); jan_szadkowski@sggw.edu.pl (J.S.); michal_drozdzek@sggw.edu.pl (M.D.); krzysztof_krajewski@sggw.edu.pl (K.K.); andrzej_radomski@sggw.edu.pl (A.R.); janusz_zawadzki@sggw.edu.pl (J.Z.); 2Institute of Chemical Technology and Engineering, Faculty of Chemical Technology, Poznan University of Technology, Berdychowo 4, 60-965 Poznan, Poland; slawomir.borysiak@put.poznan.pl

**Keywords:** bacterial cellulose gel film, enzymatic hydrolysis efficiency, liquid hot water (LHW), steam explosion (SE)

## Abstract

The influence of bacterial cellulose gel film pretreatment methods on the efficiency of enzymatic hydrolysis was investigated. An increase in the efficiency of enzymatic hydrolysis due to liquid hot water pretreatment or steam explosion was shown. The glucose yield of 88% was obtained from raw, non-purified, bacterial cellulose treated at 130 °C. The results confirm the potential of bacterial cellulose gel film as a source for liquid biofuel production.

## 1. Introduction

Fossil fuels are highly efficient energy resources, however, the same fuels contribute to an excessive burden on the environment and climate by generating huge amounts of greenhouse gases. One of the alternative sources of energy are biofuels, of which bioethanol has the greatest utility potential. Bioethanol can be successfully produced from wholesome agricultural raw materials (corn, sugar cane, potatoes, cereals) and then independently used as fuel or as a component of traditional fuels [1]. Tree biomass could be an important alternative to agricultural raw materials, however, as a result of intensive tree felling, the forest area is drastically reduced. Another interesting solution may be the use of second-generation raw materials (i.e., waste materials such as post-consumer lignocellulosic products [2,3] or waste food products [3]). Despite the easy availability of this type of waste, their use and processing into biofuel thus far is burdened with high technological and logistic costs. The search for alternative biomass for energy purposes is a necessity, and the involvement of microorganisms in the production of biofuels can be an interesting solution [4]. Bacterial cellulose synthesized by acetic fermentation microorganisms does not contain lignin, which largely inhibits the hydrolysis process, and may be an interesting raw material for the production of biofuels.

Bacterial cellulose is a natural biopolymer with very good mechanical and chemical properties. Its nanostructure, high crystallinity, flexibility, or absorption capacity make it a material with a wide range of applications [5,6,7]. The possibilities of technological use of bacterial cellulose are enormous. So far, the most numerous research has focused on the use of biocellulose in medicine [8] or food technology [9], although cellulose polymers are gaining importance in the textile [10], paper [11], electronics [12], and environmental protection industries [13]. The use of bacterial cellulose in the production of biofuels has not been the subject of numerous analyses in the literature. It seems that the main barrier limiting the use of bacterial cellulose as an alternative source for the production of biofuels is still the high cost of polymer synthesis. Although the production of biocellulose is very capital-intensive, scientists are still working on the development of new methods of reducing production costs by selecting sufficiently efficient strains, optimizing fermentation reactors, or using cheap nutrient substrates (e.g., waste agricultural raw materials). Velásquez-Rianõ and Bojacá [14] presented alternative low-cost substrates that can be successfully used in the cultivation of cellulose-synthesizing bacteria. 

The technology of converting bacterial cellulose to monosaccharides, through the use of various methods of pretreatment and then the hydrolysis process, is characterized by better efficiency than the processing of plant cellulose. This is related, inter alia, with the chemical purity of the biopolymer and a much higher degree of polymerization [15]. Although bacterial cellulose is not a pure polymer because it contains a certain percentage of additional water-soluble polysaccharides such as acetane [16] or levane [17], they do not adversely affect the efficiency of the hydrolysis process in the same way as lignin, which is a natural copolymer of plant tissues [18].

The quality of bacterial cellulose and its properties are inextricably linked to its degree of crystallinity and polymerization. The degree of polymerization of bacterial cellulose can range from 1000 to as much as 16,000–20,000 and is determined, inter alia, by the conditions and method of breeding [19,20]. Undoubtedly, another advantage that is associated with the hydrolysis of bacterial cellulose is the much lower production of by-substances that may inhibit the hydrolysis process itself, and/or that decrease the efficiency of bioethanol synthesis at a later stage. 

The enzymatic hydrolysis of cellulose to reducing sugars is catalyzed by cellulolytic enzymes. Without pretreatment of the polymer, this process is significantly less efficient due to the presence of accompanying substances, which are natural inhibitors of the enzymatic reaction. In the case of plant celluloses, these are lignin and, to a lesser extent, hemicelluloses, which make it difficult for enzymes to access the cellulose molecule [21]. Enzymatic hydrolysis is carried out by the synergistic action of several enzymes (e.g., endo-glucanases, exo-glucanases, and β-glucosidases). Endoglucanases hydrolyze regions of low crystallinity, creating the so-called free ends in the polymer; exoglucanases lead to the release of glucose dimers (cellobiose); and β-glucosidase converts cellobiose to glucose [22]. Commercial enzymatic preparations contain, apart from cellulolytic enzymes, a number of hemicellulolytic enzymes such as endo-1,4-β-D-xylanases, exo-1,4-β-D-xylosidases, endo-1,4-β-D-mannanases, β-mannosidases, acetyl xylan esterases, α-glucuronidases, α-L-arabinofuranosidases, and α-galactosidases, which hydrolyze hemicelluloses and intermediate decomposition derivatives [23].

Various pretreatment methods are used during the processing of plant cellulose to improve the efficiency of enzymatic hydrolysis. The most widely-known methods of physicochemical treatment that are favorable from the point of view of the efficiency of the enzymatic hydrolysis of plant biomass seem to be the steam explosion (SE) [24,25,26] and liquid hot water (LHW) methods [25,26]. Additionally, the SE method is characterized by low-environmental impact [27]. It therefore appears that the same pretreatment methods can be successfully used to process a bacterial cellulose gel film.

The aim of the study was to determine the efficiency of the enzymatic hydrolysis of the cellulose gel film subjected to physicochemical pretreatment using the steam explosion and liquid hot water methods. The stage preceding the pretreatment was to perform the qualitative characteristics (degree of polymerization and crystallinity) of the polymer produced on the microbiological basis. Tests were carried out to determine whether the biopolymer, not purified from the residue of the post-culture medium, may be suitable for enzymatic hydrolysis processes. 

## 2. Materials and Methods

The research used bacterial cellulose synthesized by a consortium of bacteria and yeasts called SCOBY, grown on a liquid medium containing 10% food sucrose (Krajowa Spółka Cukrowa SA, Toruń, Poland) and 0.03% peptone (Biomaxima SA, Lublin, Poland). The cultivation of the microorganisms was carried out for a period of 21 days in a laboratory incubator (J.P. Selecta Laboratory Equipment Manufacturer, Barcelona, Spain) at 26 ± 2 °C and under a relative air humidity of 66 ± 2%. The synthesized cellulose was in the form of a glassy white gel film about 10 mm thick and about 100 × 100 mm in size. After the end of the incubation time, the produced biopolymer was removed from the culture surface and divided into portions for individual analyses.

Bacterial cellulose intended for the polymerization degree and crystallinity determination was purified by rinsing in 0.1 M NaOH (POCH, Gliwice, Poland) for 30 min at 90 °C, then in distilled water, followed by 0.1% citric acid (POCH, Gliwice, Poland), and again by rinsing several times in distilled water. The cellulose prepared in this way was then dried in an oven (J.P. Selecta Laboratory Equipment Manufacturer, Barcelona, Spain) at 22 ± 2 °C to a constant mass and the polymer was obtained in the form of a film. 

The polymer intended for pretreatment by the liquid hot water (LHW) and steam explosion (SE) methods was rinsed four times in distilled water for a period of 12 h, which was a deliberate effort to show that only the surface-treated polymer could be used for further treatment, and the post-culture ingredients integrated with the polymer did not interfere with the efficiency of the enzymatic hydrolysis process. The washed polymer was stored at 4 °C.

The absolute humidity of the bacterial cellulose intended for the pretreatment methods was determined. About 1 g of the samples was taken for moisture testing from three different places of the previously rinsed polymer. The samples were dried in a laboratory oven at 103 ± 2 °C. Additionally, the dry matter content of the material before and after the pretreatment was determined.

### 2.1. Polymerization Degree of the Cellulose Samples and Crystallinity Degree

The size exclusion chromatography (SEC) technique was used to analyze the weight average polymerization degree of the cellulose samples. In the dissolution procedure of the cellulose samples, the earlier studies were used with some modifications [28,29]. The modified procedure is detailed below. The cellulose samples (15 mg) were poured into tubes with 3 cm^3^ of distilled water and left for 24 h. The next day, the samples were transferred to G3 Schott separating funnels, which were placed in vacuum suction flasks. In a further step, the water was exchanged to the less polar substances (methanol and DMAc—N,N-dimethylacetamide). At the beginning, the cellulose was washed with methanol (Chempur, Piekary Śląskie, Poland), filtered, and poured with the next portion of methanol, before being left for 1 h. The activation procedure in methanol was repeated twice. Next, the cellulose was washed thrice with DMAc (Sigma-Aldrich, Taufkirchen, Germany), and after pouring the last portion of DMAc, the cellulose was left for 24 h. Then, the cellulose was washed again with DMAc and the activated cellulose was filtered, carried to glass screw-cup tubes, and poured with 4 cm^3^ of 8% LiCl in DMAc. The mixer RM-2M (Elmi, Calabasas, CA, USA) was used in the dissolution process of the activated cellulose. After dissolution (from three to five days), 0.2 cm^3^ of the cellulose sample was diluted to a concentration of 0.5% LiCl (Sigma-Aldrich, Taufkirchen, Germany) with 3 cm^3^ of DMAc. For each sample, three analyses were performed for each sample.

The dissolved cellulose samples were analyzed using the HPLC (High-Performance Liquid Chromatography) LC-20AD system (Shimadzu, Kyoto, Japan) equipped with a differential refractive detector RID-10A, a LC-20AD pump, a DGU-20A degasser, CTO-20A oven, and a CBM-20A controller. The analysis was conducted using a PLgel column (10 µ, MIXED-B, 7.5 × 300 mm), connected to a PLgel guard column (10 µ GUARD, 7.5 × 50 mm), at the following conditions: 0.5% LiCl/DMAc as the eluent; oven temperature, 80 °C; flow rate, 1 cm^3^/min; injection volume, 0.2 cm^3^.

The HPLC system was supported by LC Solution v.1.21 SP1 software (Shimadzu, Kyoto, Japan). To process the chromatographic data, the PSS WinGPC scientific 2.74 and PSS calibration program V2.99 (Polymer Standard Service, Mainz, Germany) software were used. Narrow polystyrene standards (Agilent Technologies, Palo Alto, CA, USA) dissolved in 0.5% LiCl/DMAc were used for calibration. The molar masses of the standards used were: 6,570,000, 2,403,000, 729,500, 301,600, 117,700, 70,500, 27,810, and 9570 g/mol. The molar mass distribution of the cellulose was determined by the universal calibration method, assuming that the hydrodynamic volume of the polymer coil in solution is proportional to the product of its molar mass and intrinsic viscosity [η], while intrinsic viscosity can be calculated from the Mark–Houwink Equation (1).
 [η] = K × M^α^, (1)
where K and α are the parameters, which depend on the polymer type, solvent used, and temperature. In this work, these parameters were taken as follows: for polystyrene, K = 17.35 × 10^–3^ cm^3^/g and α = 0.642 [30] and for cellulose, K = 2.78 × 10^–3^ cm^3^/g and α = 0.957 [31]. Based on the weight average molar mass of cellulose (M_w_), the weight average polymerization degree was calculated by Equation (2):DP_w_ = M_w_/162,(2)
where 162 is the molar mass of the glucopyranose unit.

The crystallinity of thee bacterial cellulose film was analyzed using a TUR M-62 X-ray diffractometer (Carl Zeiss AG, Jena, Germany). A copper anode (Cu K_α_λ = 1.5418 Ă) at 30 kV and anode excitation at 25 mA were used. Other measurement parameters were as follows: scanning angle range—5–30°; counting stape—0.04°/3 s; experimental temperature—20 °C. Based on the diffraction patterns, the diffraction maxima were separated using the method described by Hindeleh and Johnson [32]. After separating the diffraction peaks, the amorphous area and the background, structural calculations of the value of the degree of crystallinity were performed.

### 2.2. Liquid Hot Water (LHW) and Steam Explosion (SE) Pretreatment Processes

The obtained bacterial cellulose gel film was washed four times with distilled water at room temperature (about 20 °C) before treatment. The cellulose between each experiment was kept in a plastic cuvette together with distilled water (to prevent drying) at 6 °C. For both pretreatment processes, the cellulose was divided into samples of approximately 30 × 30 mm before treatment. The material was not subjected to a drying process before pretreatment in order to counteract the closing of the porous structure, which may affect the efficiency of the enzymatic hydrolysis process [33]. From the pre-shredded material, a sample of approximately 140 g of wet material was taken and placed in a stainless steel autoclave. The volume of the high-pressure reactors in both cases was 250 cm^3^ [28]. Once the sample was placed in the reactor, distilled water was added to fill the entire volume of the reactor. The pretreatment process was carried out at 190 °C and 130 °C with a residence time of 15 min. A temperature of 190 °C was chosen based on previous studies as the most favorable temperature for poplar wood biomass [28]. The temperature of 130 °C was selected on the basis of the literature data on the resistance of the individual structural components of wood to temperature [34]. After reaching the set temperature and the LHW process time, the pressure autoclave was placed in an ice bath at approx. −5 °C. The next day, the liquid and solid fractions obtained were separated using filter paper placed in a Büchner separating funnel. The solid fraction was washed with distilled water to pH 6–7, collecting the filtrate together with the resulting liquid fraction. The quantity of the liquid fraction after LHW was obtained, together with the liquid after rinsing the solid fraction, which were collected in a volumetric flask and made up to 1 dm^3^. Two experiments were conducted for each LHW pretreatment temperature. In the case of the pretreatment with SE, once the set parameters (temperature and process time) were reached, a sudden decompression of the system was carried out by abruptly opening a ball drain valve equipped with a pneumatic motor. The material was then transferred to a receiver from which it was rinsed into a 5 dm^3^ beaker with distilled water applied under pressure. The liquid and solid fractions were separated with a Büchner separating funnel. The amount of liquid fraction obtained was measured using a large measuring cylinder. The volume of the measured liquid fraction for SE was approximately 2.5 dm^3^. Two experiments were conducted for each SE pretreatment temperature. The solid material after pretreatment was removed from the Büchner separating funnel and placed in a polyethylene bag and then stored in a laboratory refrigerator at 6 °C. A 200 cm^3^ of sample was taken from the liquid fraction after each treatment (LHW and SE) and placed in a polypropylene sealed container, which was placed in a laboratory refrigerator at 6 °C.

### 2.3. Enzymatic Hydrolysis and HPLC Sugar Analysis

The enzymatic hydrolysis was carried out on the raw bacterial cellulose and solid fraction obtained after the LHW or SE methods. The enzymatic hydrolysis process was conducted according to the method described by Antczak et al. [35] with a cellulose concentration of 1 % *w*/*w*. The sodium azide (Avantor Performance Materials Poland S.A., Gliwice, Poland) at a concentration of 2% was used as a substance preventing the growth of microorganisms during the hydrolysis. The enzymatic hydrolysis process was carried out in sealed screw-capped test tubes and the total volume of the mixture was 10 cm^3^. The Cellic CTec2 enzyme (Novozymes, Bagsvaerd, Denmark), with an activity of 148 FPU/ cm^3^ determined according to Adney and Baker [36], was used in the hydrolysis process. Samples were hydrolyzed during 72 h using a mixer (RM-2M, Elmi, Calabasas, CA, USA) with a rotational speed of 25 rpm at 50 °C. After the process, the glucose content determination in the supernatant by the HPLC method was carried out at the conditions described in the next section. All enzymatic hydrolysis tests were conducted in triplicate and single standard deviations were calculated. 

The yield of glucose produced by enzymatic hydrolysis was determined as the ratio of the increase in glucose mass, taking into consideration the mass of glucose present before the start of hydrolysis (from brew and pretreatment), to the theoretical value, obtained from cellulose mass multiplied by a 1.11 factor due to the molar mass increase from 162 to 180 g/mol when converting the glucopyranose unit to free glucose (adding the equimolar amount of water). The masses of cellulose in the samples were determined while washing the samples for the SEC and XRD analysis using Equation (3).
(3)$$\mathrm{glucose}\;\mathrm{yield}=\frac{\mathrm{total}\;\mathrm{glucose}\;\mathrm{mass}\;/g-\mathrm{glucose}\;\mathrm{mass}\;\mathrm{before}\;\mathrm{hydrolysis}\;/g}{1.11\times \mathrm{cellulose}\;\mathrm{mass}\;/g\;}\times 100\%$$

### 2.4. Assay of Released Sugars and Inhibitors 

Sugars (glucose and fructose) and inhibitors (formic acid, levulinic acid, and 5-hydroxymethylfurfural—HMF) in the liquid fraction obtained after the LHW and SE pretreatments were determined by HPLC (LC-20AD system, described above). The chromatographic analysis conditions were as follows: a RHM-Monosaccharide (300 mm × 7.80 mm, Rezex, Torrance, CA, USA) column, running at a flow rate of 0.6 cm^3^/min^−1^ at 80 °C, with redistilled water as the eluent.

## 3. Results and Discussion

### 3.1. The Degree of Polymerization and Crystallinity of the Bacterial Cellulose

Crystallinity and the degree of polymerization are important factors influencing the quality of cellulose. In particular, the determination of the degree of polymerization is an important indicator of the quality of a biopolymer when considering it as an alternative raw material in applications for the production of biofuels. Determining the degree of cellulose polymerization before pretreatment seems to be the key determinant in assessing the efficiency of the enzymatic hydrolysis of cellulose [37]. 

On the basis of the conducted analysis, it was found that bacterial cellulose was characterized by a much higher degree of polymerization than plant cellulose (Table 1). According to Antczak et al. [38] the degree of polymerization of cellulose isolated from a 90-year-old pine, is about 2000, therefore, the obtained bacterial cellulose was characterized by a three times higher degree of polymerization than its plant equivalent. Molar mass distribution for the exemplary sample of bacterial cellulose dry film is shown in Figure 1a.

The diffraction pattern shown in Figure 1b describes the characteristic of the cellulose-derived diffraction maxima (cellulose polymorph I) at 2θ diffraction angles of 15, 17, and 22.5°. The mean value of the crystallinity degree was 46% (Table 1), which was a lower value than the average crystallinity for bacterial cellulose obtained on the standard Hestrin and Schramm medium [39]. Based on the literature data, it can be concluded that the polymer crystallinity degree results from the hydrogen bonds between the chains, and therefore different types of the culture medium, the cultivation method, and the types of microorganisms used may affect the morphological features of the bacterial cellulose [40]. The degree of crystallinity of cellulose is a determinant of the physical and mechanical properties of the polymer including the reactivity of cellulose in the enzymatic hydrolysis process, while lower crystallinity promotes the enzyme action on the cellulose chain [41].

### 3.2. Properties of Bacterial Cellulose Gel Film after the LHW and SE Pretreatments

The raw bacterial cellulose gel film was characterized by very high humidity, which is a natural feature of the polymer. Bacterial cellulose is a hydrophilic polymer, and the hydration humidity is associated with the presence of numerous -OH groups, which can form inter- and intra-molecular hydrogen bonds, but also depends on the microstructure of the polymer [42,43]. During the SE pretreatment, the moisture of the cellulose was reduced almost three times compared to that of the raw cellulose. This result may indicate a greater permeability of the treated polymer through a sudden explosion of water vapor. This result was confirmed by the studies carried out by Antczak et al. [28], which showed an increase in the cellulose porosity as a result of pretreatment with the SE method. The pretreatment of the bacterial cellulose gel film with hot water did not cause such a rapid change in the moisture content of the polymer, although a greater loss of moisture was observed during the LHW treatment at higher temperatures. Studies by Kashcheyev et al. [41] confirm that cellulose moisture may have an influence on the rate of enzymatic hydrolysis. The same authors of the study showed a decrease in the rate of the enzymatic reaction in the case of dry polymer hydrolysis, with no effect of polymer moisture on the final yield of the reducing sugars. 

The percentage content of dried biomass is shown in Table 2. Based on the obtained results, it can be concluded that the dry matter content of the polymer was relatively low, and the polymer itself was characterized by a very high hygroscopicity. There was a higher percentage of the dry matter of bacterial cellulose pre-treated with the SE method than after the LHW method. 

### 3.3. Enzymatic Hydrolysis and HPLC Sugar Analysis

The analysis of the yield of reducing sugars as products of the enzymatic hydrolysis reaction was carried out for the raw and pretreated cellulose using the LHW and SE methods. The results of the analysis are presented in Table 3. Based on the tests performed, it should be stated that the temperature of the pretreatment process had a decisive influence on the efficiency of enzymatic hydrolysis. The average glucose yield after bacterial cellulose gel film hydrolysis, preceded by LHW and SE pretreatment at 130 °C, was very similar and approximately 88%. The enzymatic hydrolysis efficiency of the polymer, subjected to the same pretreatment processes (LHW and SE), but at a higher temperature (190 °C), was clearly lower and amounted to 80.1% and 67.3%, respectively. The average yield of glucose in the enzymatic hydrolysis of raw cellulose was 58.4%. 

In similar studies by Kashcheyev et al. [41], the hydrolysis efficiency of the crude and untreated bacterial cellulose was 56.3–66.6%, thus similar to the values obtained in this study for raw cellulose. It should also be stated that the hydrolysis efficiency of the non-purified pretreated bacterial cellulose significantly exceeded the hydrolysis efficiency of the plant biomass [35,44,45]. In studies conducted by Antczak et al. [28], the glucose yield from the enzymatic hydrolysis of cellulose isolated from *P. trichocarpa* wood subjected to the LHW and SE pretreatment methods was up to 67%. Therefore, it should be stated that bacterial cellulose seems to be a good material, being an alternative source of raw materials suitable for the production of biofuels. The reduced amount of glucose formed at higher temperatures of the presented LHW and SE physicochemical treatments may also result from the formation of by-products, mainly furan compounds and organic acids [28]. This problem may be especially important for the enzymatic hydrolysis of the solid fraction obtained after the SE pretreatment because in this process, the unwashed material was used.

### 3.4. Released Sugars and Inhibitors

The methods of pretreating bacterial cellulose can generate the formation of inhibitors that negatively affect the efficiency of the hydrolysis process. Zawadzki et al. [46] found that the presence of furfural in the liquid fraction of poplar wood pretreatment reduced the activity of the enzymatic reaction of the Dyadic Cellulase CP CONC enzyme complex.

The analyses of the liquid fraction remaining after pretreatment of the bacterial cellulose gel film by the LHW and SE methods are shown in Figure 2a,b. Based on the results presented in Figure 2a, it was found that the sum of the released sugars depended on the temperature of the process. The average sum of sugars released after pretreatment of the polymer at 190 °C was 0.6% (SE) and 1.4% (LHW), respectively, and was, respectively, 4.5 and 12 times lower than the average sum of sugars released to the liquid fraction during the same treatment processes, but carried out at a lower temperature. At a higher temperature of the treatment processes, a higher average sum of formed inhibitors was observed (Figure 2b). Similar relationships were obtained by Akus-Szylberg et al. [47] for plant cellulose. Moreover, it is very interesting that as a result of the pretreatment of bacterial cellulose, the total concentration of inhibitors was at a much lower level than the analogical obtained results of the pretreatment of plant biomass. According to Antczak et al. [28], the sum of inhibitors after 15 min of the steam explosion process at 190 °C of *P. trichocarpa* wood was 22.1%, while for the liquid hot water pretreatment in the same conditions, it was 13.7%. In addition, some fructose was observed in the liquid fraction, which is probably residue contamination from the medium broth found in the bacterial cellulose gel film structure. Moreover, soluble oligosaccharides resulting from the degradation of cellulose may also be present in the liquid fraction. However, in the HPLC system used, it was not possible to separate them and determine their content.

Levulinic acid dominated among the inhibitors generated in the SE and LHW processing. The percent concentrations of levulinic acid after the SE and LHW pretreatments were between 0.85% and 2.80%. 5-Hydroxymethylfurfural (HMF) was found in the liquid fraction obtained after the SE and LHW pretreatments of cellulose at 190 °C. The percent concentrations of HMF generated in these conditions were 0.15% and 0.40%, respectively. HMF was also found at 130 °C after the SE and LHW pretreatments of cellulose (0.01% and 0.29%, respectively).

During the pretreatment processes at 190 °C, an appearance of formic acid was observed (1.43%, 0.03%, after SE and LHW, respectively), which was not observed at the lower temperature of the pretreatment process at 130 °C. According to Arora et al. [48] and Jönsson et al. [49], formic acid is one of the most potent inhibitors of cellulose enzymatic hydrolysis. It can therefore be assumed that the lower values of the enzymatic hydrolysis efficiency of bacterial cellulose may be related to the appearance of formic acid in the liquid fraction. Formic acid is one of the dominant inhibitors in the liquid fraction, formed after the pretreatment of bacterial cellulose by the SE method at 190 °C, and the hydrolysis of the solid fraction after this pretreatment process turned out to be the least efficient.

## 4. Conclusions

The results obtained in this study, concerning the effect of the pretreatment methods on the efficiency of enzymatic hydrolysis, showed that both the type of pretreatment and the temperature of the process had an influence on the efficiency of glucose obtained during the enzymatic hydrolysis of the bacterial cellulose gel film. Higher process temperatures generated greater amounts of inhibitors that may inhibit the enzymatic activity. In the pretreatment processes carried out at 190 °C, the presence of formic acid in the liquid fraction, which is a strong inhibitor of hydrolytic enzymes, was observed. Its presence may result in a lower glucose yield at elevated temperatures. At the lower temperature of the cellulose pretreatment, more glucose was observed, which passed into the liquid fraction. In this fraction, fructose was also identified as an impurity in the cellulose polymer, derived from the breakdown of sucrose, which was the main component of the culture medium.

The pretreatment of the bacterial cellulose gel film by the LHW and SE methods at 130 °C allowed us to achieve good hydrolysis results. This may be of great importance in the case of the practical design of the bacterial cellulose process for specific applications (e.g., bioethanol production).

## Figures and Tables

**Figure 1 polymers-14-02032-f001:**
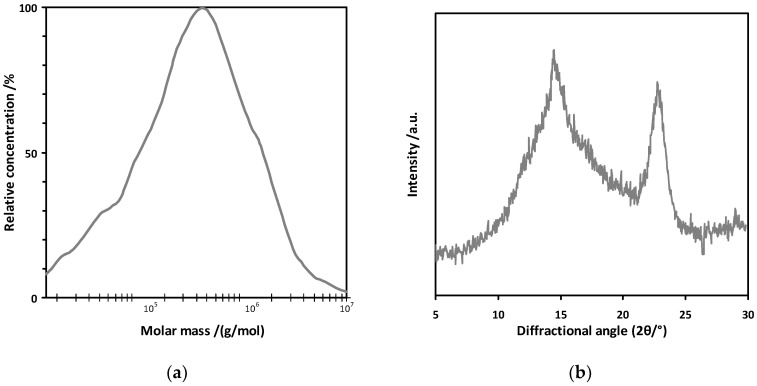
The raw bacterial cellulose properties: (**a**) molar mass distribution; (**b**) XRD curve for crystallinity determination.

**Figure 2 polymers-14-02032-f002:**
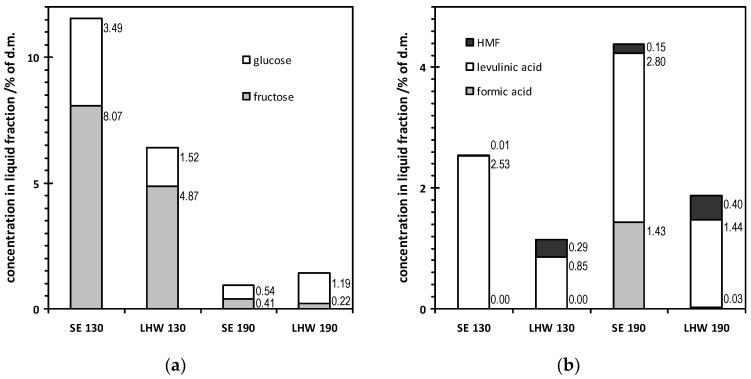
Composition of the liquid fraction obtained after the SE and LHW pretreatments at 130 °C and 190 °C as a percentage of soluble dry mass: (**a**) simple sugars; (**b**) inhibitors.

**Table 1 polymers-14-02032-t001:** Properties of the raw bacterial cellulose film.

Parameter	Value (SD)	Notes
Crystallinity degree	46 (0.8) %	dry film
Molar mass: Number average M_n_ Weight average M_w_ Molar mass dispersity Đ Polymerization degree DP_w_	324 (73) kg/mol 984 (60) kg/mol 3.2 (0.9) 6080 (370)	dry film

**Table 2 polymers-14-02032-t002:** Properties of the treated bacterial cellulose.

Pretreatment Method	Dry Matter Content of Bacterial Cellulose (SD)
Raw cellulose LHW, 130 °C	2.8 (0.4)% 3.0 (0.7)%
LHW, 190 °C	3.9 (0.6)%
SE, 130 °C	5.4 (0.5)%
SE, 190 °C	7.8 (0.16)%

**Table 3 polymers-14-02032-t003:** Cellulose to glucose hydrolysis yield.

Pretreatment Method	Glucose Yield (SD)
Raw (non-treated)	58.4 (3.4)%
LHW, 130 °C	88.0 (2.4)%
LHW, 190 °C	80.1 (1.5)%
SE, 130 °C	88.7 (1.8)%
SE, 190 °C	67.3 (2.7)%

## Data Availability

Not applicable.

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
