# Peer review of "Evaluation of the Hydrolysis Efficiency of Bacterial Cellulose Gel Film after the Liquid Hot Water and Steam Explosion Pretreatments"

_polymers, 2022, doi:10.3390/polym14102032_

Round 1

Reviewer 1 Report

In this study, the authors pretreated bacterial cellulose gel film using liquid hot water and steam explosion methods and investigated the effectiveness of its conversion to degraded sugars. The manuscript was written in an understandable way, however it presents a few issues that may require the authors to further address.

  1. The introduction may be further polished, specifically for the first paragraph. it sounds too verbose.
  2. Line 53-54, bacterial cellulose does not contain 100% cellulose component. The percentage of cellulose may account for 90-95%, others may have proteins, lipids, and some extracellular polysaccharides. These extracellular polysaccharides may have similar structures to hemicelluloses. The literature has published relevant articles in these topic. 
  3. The authors focus on the enzymatic conversion of cellulose. Although this is not major topic in this study, it is suggested for the authors to add some background information about enzymatic conversion of cellulose into biofuels, including the use of commercial enzymes and treatment process.
  4. The subtitles under 3. Results and Discussion should be renumbered. 
  5. The description of the inhibitors was not clear. How was the inhibitors generated during the enzymatic degradation process? How did these inhibitors affect the conversion of cellulose to glucose/fructose? Did the cellulose degradation only produce glucose and fructose without any trace of other oligocelluloses?
  6. The figure 1 and 2 should be switched.

Reviewer 2 Report

In this paper, the authors evaluated the efficiency of enzymatic hydrolysis of cellulose gel film subjected to physicochemical pretreatment using the steam explosion and liquid hot water methods. According to the results, higher process temperatures can inhibit enzymatic activity. At the lower temperature of the cellulose pretreatment, more glucose is observed which passed into the liquid fraction. As a result, the low pretreatment temperature of bacterial cellulose gel film by the LHW and SE methods favors specific applications including bioethanol production. The paper is well presented, data was solid, and the investigation is of interest to researchers in related fields. And thus, I recommend the publication of this paper in Polymers.

Round 2

Reviewer 1 Report

N/A